# Structural models for spreading and scaling digital health initiatives: A scoping review protocol

Celia Laur[1,2]*, Karen Lee[3], Andrew Milat[3,4], Samuel Petrie[5,6], Zeenat Ladak[1,2], Vincci Lui[2], Alix Hall[7,8,9,10], Nicole Nathan[7,8,9,10], Priscilla Medeiros[11], Noah Ivers[1,2,12], Onil Bhattacharyya[1,12]

1 Office of Spread and Scale, Women's College Hospital Institute for Health System Solutions and Virtual Care, Toronto, Ontario, Canada, 2 Gerstein Science Information Centre, University of Toronto, Toronto, Ontario, Canada, 3 Prevention Research Collaboration, Charles Perkins Centre, Faculty of Medicine and Health, Sydney School of Public Health, The University of Sydney, Camperdown, Australia, 4 New South Wales Agency for Clinical Innovation, St Leonards, Australia, 5 Implementation Science Team, Research, Innovation, and Discovery, Nova Scotia Health, Halifax, Canada, 6 Department of Community Health and Epidemiology, Faculty of Medicine, Dalhousie University, Halifax, Canada, 7 School of Medicine and Public Health, Faculty of Health and Medicine, University of Newcastle, Newcastle, New South Wales, Australia, 8 The National Centre of Implementation Science (NCOIS), The University of Newcastle, Newcastle, New South Wales, Australia, 9 Hunter Medical Research Institute, New Lambton Heights, Newcastle, New South Wales, Australia, 10 Hunter New England Population Health, Hunter New England Local Health District, Wallsend, New South Wales, Australia, 11 Edwin S.H. Leong Centre for Healthy Children, University of Toronto, Toronto, Ontario, Canada, 12 Department of Family and Community Medicine, University of Toronto, Toronto, Canada

* celia.laur@wchospital.ca

## Abstract

### Background

Healthcare initiatives have a larger impact if effective initiatives are spread (brought from one site to the next) or scaled (infrastructure developed to underpin and support widespread implementation), while sustaining initial benefits. Unfortunately, many initiatives, including digital health initiatives, remain confined to the pilot stage. Of those initiatives that do progress, little is known about how to plan for the equitable spread and scale of effective initiatives. There are many structural "models" of spread and scale, defined here as conceptual representations of how initiatives are organised and delivered across multiple settings (i.e., hub-and-spoke model), yet little is known about these models.

### Objectives

*Primary Objective:* To identify and describe structural models for spreading and scaling digital health initiatives. *Secondary Objectives:* 1. To describe the associated factors, strengths, limitations, and necessary preconditions associated with each model. 2. To describe the barriers and facilitators experienced when applying each model. 3. To explore whether and how each model prioritized equitable delivery of care. 4. To

**Data availability statement:** No datasets were generated or analyzed during the current study. All relevant data from this study will be made available upon study completion.

**Funding:** The author(s) received no specific funding for this work.

**Competing interests:** The authors have declared that no competing interests exist.

**Abbreviations:** CFIR 2.0: Consolidated Framework for Implementation Research 2.0, JBI: Joanna Briggs Institute (JBI), NASSS: Non-adoption, Abandonment, Scale-up, Spread, and Sustainability framework

determine which pre-established types of scale (horizontal, vertical, diversification, and spontaneous) are associated with each model.

## Methods

A scoping review will be conducted following Joanna Briggs Institute (JBI) methodology and reported in accordance with PRISMA-ScR guidelines. The search strategy includes peer-reviewed databases for health and business, alongside grey literature sources. Eligibility criteria follow the Population-Concept-Context framework, focusing on digital health initiatives delivered in healthcare settings.

## Results

The review will produce a comprehensive overview of structural models for spreading and scaling digital health initiatives, including model names, descriptions, strengths, limitations, preconditions, associated barriers and facilitators of applying each model, relationships between models and established types of scale, and equity considerations.

## Conclusions

This novel review aims to inform practical planning of how to bring digital health initiatives to new settings and populations, to support more equitable access to these initiatives.

---

## Introduction

Healthcare initiatives are more likely to achieve population-wide benefits in line with the Quintuple Aim (patient experience, health outcomes, reducing costs, care team well-being, and health equity) [1,2] if the initiative is spread (brought from one site to the next), or scaled (have the infrastructure that supports delivery across a wide region, such as a province or country), [3] effectively. This can mean that benefits seen during initial implementation are maintained, ideally with increased efficiency. [4] This is also the case for digital health initiatives, which hold promise for improving patient outcomes and service efficiency, yet many effective initiatives often fail to move beyond the pilot stage. [5–10] Simultaneously, millions of dollars are spent to scale initiatives with insufficient evidence of effectiveness, in order to meet political and strategic priorities. [4,11,12] When initiatives are spread and scaled, it can be opportunistic or unplanned, which ultimately benefits those most able to support the change, while those most in need of the initiative are typically the least likely to benefit. [13] This is particularly evident in digital health initiatives, which can emerge rapidly, and may need to be rolled out rapidly in order to achieve the desired impact. The lack of strategic planning of how to spread and scale effective innovations can increase inequitable access to care, leaving those who are underserved behind. [14] For example, well-resourced organizations may be better positioned to leverage

spontaneous opportunities for improving patient care or programs designed for urban settings may be introduced to rural areas without sufficient attention to what adaptations are required to meet context-specific needs when the spread and scale occurs spontaneously compared to with more deliberate planning. (i.e., including community consultations, needs assessments) [14–19].

The field of *implementation science* studies methods and strategies that support the uptake of evidence-informed practice and research into regular practice, including for initial implementation, sustainability, spread, and scale [20]. Thus far, implementation science has focused more on studying initial implementation, yet different approaches are needed for scale. [21] There is limited evidence about how health systems should plan for spread and scale, particularly when we consider the complexity of making system-level changes. [3,6,12,22–27] While often used interchangeably and/or inconsistently [12,23], for this work, "*spread*" is interpreted as being brought from one site to the next (i.e., one hospital to another), while "*scale*" is defined as developing infrastructure to underpin and support widespread implementation of initiatives across jurisdictions, contexts or settings. [3] Decisions to scale can be complex, non-linear, influenced by political and strategic needs, driven by demand, and generally require institutional or governmental support, such as infrastructure, policies, and system-level change [3,21].

Digital health can be defined as "*the use of information and communication technologies in medicine and other health professions to manage illnesses and health risks and to promote wellness".* [28] Examples of digital health include, but are not limited to, telemedicine or virtual care, mobile health, wearables, remote patient monitoring, electronic health records or patient portals, and artificial intelligence. Several frameworks and tools have been developed to support the spreading and scaling of public health and healthcare initiatives, [27] including for digital health. [29,30] For example, the Non-adoption, Abandonment, Scale-up, Spread, and Sustainability (NASSS) framework was developed to enhance the ability to predict and assess the success of implementing innovative technologies in healthcare. [29,30] The NASSS acknowledges the inherent complexity of scaling, and is a useful tool to support planning and evaluation, [29,30] which could be used alongside other approaches including those explored in this protocol.

The *ExpandNet Scaling-up Framework,* the most frequently used framework for scale planning in healthcare [12], aims to systematically analyze and support necessary actions for sustainable scale. [31] Within the ExpandNet Framework, one of the strategic choice areas is "Types of Scale," three of which are deliberate: "vertical" (embedding the initiatives in policies, structures, and operational guidelines), "horizontal" (where initiatives are expanded to different geographies or to serve different populations), and "diversification" (testing and adding new interventions to an existing package). The fourth type, "spontaneous", occurs without deliberate guidance. [31] These frameworks, tools, and the Types of Scale are useful for supporting scale planning, however, there appears to be more specific structural "models" emerging that describe how spread and scale are operationalized in healthcare service settings. Considering these more structural models alongside Types of Scale, and frameworks such as NASSS, may provide more detail to help design and implement a scaling strategy, and raising more specific questions that teams will need to consider.

A model is defined as "*a description or analogy used to help visualize something (such as an atom) that cannot be directly observed".* [32] In implementation science, in the seminal paper on implementation theories, models, and frameworks, models are only described as "process models", which aim "to describe and/or guide the process of translating research into practice". [33] This review hypothesizes that there are other types of structural models when spreading and scaling healthcare initiatives, similar to models more commonly applied in commercial sectors. We are defining these models for spreading and scaling as conceptual representations of how initiatives are organised and delivered across multiple settings. Models of spread and scale aim to demonstrate underlying structure, which differentiates them from process models which focus on the steps needed to achieve that structure. For example, a remote monitoring program can spread and scale using a "hub and spoke" model, where a central "hub" coordinates and supports the main infrastructure for monitoring, while the "spokes" could be more region specific focusing on recruitment of individuals to be monitored, adapting service offerings based on the needs of that region. [34,35] A process model may be useful for implementing the structural models for spread and scale. Currently, there is no known list of

these structural models, no consolidated definitions, and limited guidance on when and how each model could be used. Due to the proliferation of digital healthcare initiatives following the COVID-19 pandemic, and the penetration of digital health programs across health system structures and organization, they provide an ideal opportunity to begin to explore these models [36–41].

The objectives of this review include:

*Primary Objective*: To identify structural models for spreading and scaling digital health initiatives and describe how they are used.

*Secondary Objectives:*

1. To describe and compare the associated factors, strengths, limitations and necessary preconditions associated with selecting each model.

2. To describe the barriers and facilitators experienced when applying each model.

3. To explore whether and how each model prioritized equitable delivery of care.

4. To determine which pre-established types of scale (horizontal, vertical, diversification, and spontaneous) are associated with each model.

## Methods

### Study design

A scoping review will be conducted following guidelines developed by the Joanna Briggs Institute (JBI) and the review will be reported in line with the PRISMA-ScR (Preferred Reporting Items for Systematic Reviews and Meta-Analyses Extension for Scoping Reviews), which are best-practice reporting guidelines for scoping reviews.

The review was registered to Open Science Framework on October 27, 2025.

Ongoing discussions will be held with the wider research team (authors on this protocol, and additional team members to support screening) throughout analysis of results. The team includes spread and scale experts (CL, AM, KL, OB) health services researchers (CL, ZL, NI, AH, NI, SP, OB), clinicians (NI, OB), implementation scientists (CL, ZL, NI, AH, NN), policy experts (AM, SP, NI, OB), and experts in review methodologies (NN, AH, KL, VL).

### Eligibility criteria

In line with JBI methodology, we will follow the Population-Concept-Context (PCC) Framework:

- *Population:* All populations (children and adults) receiving or delivering a digital health initiative.

- *Concept:* Structural models for spreading and scaling digital healthcare service initiatives (programs, services, initiatives, interventions etc.)

- *Context:* Healthcare service delivery settings (hospitals, primary care, etc.)

Detailed inclusion and exclusion criteria are provided in Table 1. A cutoff date of 2003 was chosen as this was the launch of the ExpandNet group, which increased focus on studying the science of scaling [31].

### Conceptual framework

Data extraction and analysis will follow implementation science and equity focused concepts specific to digital health:

*Digital Health Initiative*: Digital health will be categorized as: telemedicine or virtual care, mobile health, remote patient monitoring, electronic health records or patient portals, machine learning/artificial intelligence, wearables, administrative support, and "other".

**Table 1. Inclusion and exclusion criteria.**

| Inclusion | Exclusion |
|---|---|
| All study designs, publication-types (trials, qualitative interviews, case studies, commentaries, reviews etc.) and methodologies (qualitative, quantitative, and mixed methods) | Abstracts without an accompanying article |
| Focus on a digital health initiative (remote monitoring, video visit etc.). Can include administrative tools (AI scribe, automated billing etc.) | Does not focus on a digital health initiative |
| Primary focus on bringing healthcare initiative(s) (programs, interventions, tools etc.) to a new setting or population (spreading and/or scaling) | Does not have a primary focus on bringing healthcare initiative(s) (programs, interventions, tools etc.) to a new setting or population (spreading or scaling) |
| Focus on initiatives delivered in specific healthcare focused settings (hospital, primary care, long-term care etc.); | Focus on initiatives exclusively delivered in non-healthcare settings (schools, daycares, prisons etc.) |
| For intervention studies, interventions that have been spread to two or more sites | For intervention studies, interventions that have *not* spread to two or more sites |
| Published after 2003 | Published before 2003 |
| Published in English | Not published in English |

*Structural Models for Spread and Scale*: For this work, we define structural models for spreading and scaling as conceptual representations of how initiatives are organised and delivered across multiple settings. Preliminary work has identified three known models, including hub-and-spoke, franchise, and platform models. A draft list has been developed of potential other models to consider. Screeners and extractors are familiar with this information and how what we are looking for differs from theories, process models, or implementation frameworks. Following data extraction, secondary reviewers will confirm model names and re-code included studies with the review-finalized names during data synthesis.

*Barriers and facilitators*: The Consolidated Framework for Implementation Research 2.0 (CFIR 2.0), a determinant framework, will be used to name and categorize barriers and facilitators associated with each model, and highlight similarities and differences between models. The CFIR 2.0 includes five domains (inner setting, outer setting, innovation characteristics, individual, implementation process), 48 constructs with 19 subconstructs (i.e., policies and laws; partnerships and connections; external pressures) [42].

*Equity Considerations*: The Health Equity Implementation Framework is a determinants framework which aligns with the five domains of CFIR to support a deeper understanding of healthcare disparity implementation challenges. [18,43] This framework will be used to describe equity considerations specific to each model (i.e., if/how adapted to support specific populations, community engagement in scale planning, ways to support more marginalized populations to access the initiative etc.).

## Literature search

We will perform a comprehensive search for published peer-reviewed articles and a separate grey literature search.

The comprehensive search for literature will be developed by an academic health science librarian (VL). The search strategies will be based on the core search concepts: (1) spread and scale, (2) digital health, and (3) healthcare delivery and models.

A preliminary search was developed and executed in OVID Medline ALL (1946-present). Additional databases that will be searched are OVID Embase (1947-present), OVID PsycINFO (1806-present), EBSCO CINAHL Plus with Full Text (1937-present), Proquest ABI/INFORM Collection (1971-current), Wiley Cochrane Library, and Web of Science (1900-present). The initial Medline search strategy was developed based on suggested keywords from the project leads. The search strategy includes a combination of subject headings and keywords and will be translated into each database using combinations of each database platform's command language, controlled vocabulary, and appropriate search fields.

The spread and scale concept was partly developed based on adaptations from previous review protocols on scaling. [44,45] The terms used for the digital health concept were adapted from two filters on artificial intelligence [46] and telemedicine. [47] The terms used for the healthcare-related concept were assessed for sensitivity and specificity, and the concept was intentionally left fairly broad. Piloting of adding detailed criteria to this concept (listing individual healthcare settings etc.) created an overly sensitive search with a very high number of irrelevant results. The decision was made to focus on specific settings within title/abstract screening. Limits imposed are for articles published from 2003-present, and a filter for human-only studies will be applied to the Medline and Embase searches. No study design or language limits will be imposed. *Search Strategy Validation:* The Medline search strategy was validated through the retrieval of a key set of relevant studies previously identified by team members [36,44,45,48–56], and was peer reviewed by a second librarian. This search can be found in Appendix 1.

*Grey Literature Search*: The secondary grey literature search will include two phases. The first phase will include a more general search using Google, and Google Scholar, followed by a more targeted search including Overton, a large global policy database, [57] Policy Commons, the WHO Digital Library, health agencies' reports, Industry white papers, and other sites, such as the websites for ExpandNet [31] and Spring Impact. [58] Google Custom Searchers will be used to search for Canadian government sites, [59] and Canadian health agencies. [60] We will also use the approach recommended by the Canadian Drug Agency for systematic searches of grey literature [61].

As a final confirmation, we will search reference lists of included publications and consult experts in the field, including authors on this paper, to identify additional relevant studies.

## Screening

Screening will be conducted in Covidence systematic review software by a team of reviewers. A screening decision tree will be developed and piloted prior to starting full title and abstract screening, and again before full text screening. Upon completion of the search, duplicates will be removed using Covidence. Title, abstract, and authors will be visible to all reviewers; no blinding will occur.

*Title and Abstract Screening*: To pilot the title and abstract screening process, the screening team will each screen the same 10 titles and abstracts. Comparison will then be made between the reviewers, with discrepancies discussed. Reviewer instructions will be updated based on these discussions. A new set of 50 articles will then be screened, discrepancies discussed, and instructions updated. If there are 10 or more discrepancies (20%+), an additional 50 articles will be screened. If there are less than 10 discrepancies, reviewers will continue screening the remainder of the articles. All title and abstract screening will be conducted independently in duplicate, with discrepancies discussed between the team of reviewers initially, with additional team members brought in as needed.

*Full Text Screening*: For full text screening, a similar process will be taken as for title and abstract screening, however only five full texts will be reviewed in each round. All full text screening will be conducted independently in duplicate, with discrepancies discussed with the review team. When consensus cannot be reached, additional team members (non-screeners) will be consulted.

## Data extraction and analysis

All eligible full text articles will be extracted in duplicate. No extractor blinding will occur. All data will be extracted to a structured spreadsheet, with qualitative analysis conducted using qualitative analysis software. Analysis is anticipated to be an iterative and complex process, and thus a collaborative, rather than individualistic, approach will be taken.

A structured spreadsheet and guidance document will be developed for extraction. Preliminary extraction plans are provided in Table 2.

*Pilot Extraction*: To pilot the extraction process, three randomly selected articles eligible for full text extraction will be selected. All extractors will extract the same articles using the spreadsheet template. Responses will be compared,

**Table 2. Draft extraction and analysis plan.**

| Section | Data to be Extracted | Analysis Plan |
|---|---|---|
| 1: Metadata | • First author<br>• Year of publication<br>• Country of corresponding author<br>• Type of publication<br>• Patient population<br>• Patient population issue (chronic disease etc.)<br>• Initiative type (virtual care, remote monitoring etc.)<br>• Setting (Hospital, Primary Care, Long Term Care, Etc.)<br>• Definition of spread<br>• Definition of scale | Results summarized descriptively. |
| 2: Intervention Details (Intervention Studies Only) | • Country of Intervention<br>• Study design<br>• Intervention Name<br>• Intervention Origin<br>• Research<br>• Clinical Practice<br>• Other<br>• # sites initially<br>• # sites end<br>• Years of implementation (total) | Details of the intervention being scaled will be presented descriptively by setting, groupings of healthcare considerations, groupings of populations, durations, number of sites, and initiative type. |
| 3: Structural Models of Spread and Scale | • Name for the model (if provided; "literature derived")<br>• Author-Derived (suggested by extractor) name for the model (can be listed as "unclear")<br>• Description of the model or general spreading or scaling process. If multiple models are used simultaneously, each model will be extracted separately.<br>• Note of if the model is generic, or specific to a clinical topic or setting<br>• Scale Type(s) associated with the model (Vertical, Horizontal, Diversification, Spontaneous; unclear)<br>• If a visual representation is provided of the scaling model, this image will be extracted. | A list will be developed of models named directly in the included studies ("literature derived" models) Derived model names will be confirmed by two reviewers ("author-derived" models). Corresponding scale types will be confirmed by two reviewers. A final list of model names and descriptions will be developed. A visual representation will be provided of each model, if possible. |
| 4: Strengths, Limitations & Preconditions | Descriptions of conditions or considerations reported to lead to the selection of the model(s) and the factors associated with success. For example, the need to keep core operations led to the decision to follow a hub and spoke model, or lack of funding that led to more spontaneous scaling. | Content analysis of the justification for selecting the model. |
| 5: Barriers & Facilitators to implementation | Reported barriers and facilitators to the application of the model. For example, communications challenges were difficult when there were more spokes that needed to connect to a centralized hub. | Deductive analysis to the CFIR 2.0. |
| 6: Equity Considerations | Any mention of how health equity (equitable access to care, focus on specific populations, etc.) is considered in each model. | Deductive analysis to the Health Equity Implementation Framework |

discrepancies discussed, and the spreadsheet updated. If needed, an additional three articles will be extracted by the group.

*Risk of Bias and Critical Appraisal*: Risk of bias assessments will not be conducted as we are looking to generate model names and comprehensive descriptions rather than assessing the quality of the publications. Following the JBI methodology, we do not plan to critically appraise the quality of studies, as we aim to map organizational structures and identify key concepts and their descriptions, and will not be assessing effectiveness of each study or model.

*Extraction Merging*: Upon completion of double extraction, when there is divergent extraction within Metadata and Intervention details (sections 1 and 2 of the extraction and analysis plan in Table 2), the individual merging the files will check the original article to make the final decision or bring to the extractor team for discussion. For all other sections, all

responses will be kept as there may be different interpretations regarding what is relevant for the scaling process. The focus will be on comprehensiveness of data extraction rather than making initial, potentially subjective, decisions during this extraction phase. No tests will be used to assess extractor agreement as much of the text will be qualitative and both responses will be kept, thus minimizing the need for full agreement.

*Metadata and Intervention Details (Section 1 & 2 in Table 2)*: Metadata and intervention details will be summarized descriptively. Definitions of spread and scale will be grouped by definition and original reference. Raw data of intervention details will undergo data transformation including:

- Digital health initiative types will be grouped (virtual care, remote monitoring etc.)

- Settings will be combined into wider groupings as needed (hospital, primary care etc.)

- Healthcare considerations will be grouped (chronic condition, prevention etc.)

- Population type will be grouped (adult, child, older adult etc.)

*Model Name Confirmation (section 3 in Table 2)*: If the name of the model of scale is provided (i.e., hub and spoke), the names will be used to form a list of named models (literature-derived models). If more than one model name is provided, all will be extracted, acknowledging one initiative may be using multiple models. When a model name is suggested by extractors (author-derived names) or listed as unclear, two reviewers with expertise in scaling terminology, will review each paper individually and suggest a model name or multiple model names if consensus cannot be reached at this stage. The two lists will then be compared and the combined list sent to all reviewers for feedback, followed by review by the wider team. Consultation meetings will also be held to confirm the model names, preliminary descriptions, and visual representations. Studies where model names are still unclear after the second round of review will be kept separate, then reassessed after the list is finalized. Any studies that still do not fit within the list will be discussed with all reviewers and the wider team. Model names and descriptions will be finalized before progressing with additional analysis, so finalized names can be assigned to each study to support analysis by model. The final models will mention if the model name is literature derived or author derived.

*Strengths, Limitations & Preconditions (section 4 in Table 2)*: Once model names have been assigned to each study, publications will be grouped by model name, with some studies included with more than one model. Content analysis will then be used to analyse descriptions of all conditions or considerations that led to the selection of each model for scale, alongside associated factors, strengths and limitations of each model. The final product will include a list of strengths, limitations, and preconditions associated with success of each model, to help inform decisions on which model to choose.

*Barriers and Facilitators to Implementation (section 5 in Table 2)*: The Consolidated Framework for Implementation Research 2.0 (CFIR 2.0) will be used to name and categorize barriers and facilitators to the implementation of each model. [42] Initially, all barriers and facilitators will be extracted together. The first round of analysis will then group barriers and facilitators into CFIR 2.0 domains (inner setting, outer setting, innovation characteristics, individual, process) within each finalized model. The final analysis will aim to deductively map barriers and facilitators to CFIR 2.0 constructs (tension for change, compatibility etc.). [42] The final product will include a list of barriers and facilitators to application of each model at each CFIR domain.

*Equity Considerations (section 6 in Table 2)*: Deductive analysis will be conducted using the Health Equity Implementation Framework [18,43] to describe equity considerations specific to each model (i.e., if/how adapted to support specific populations, community engagement in scale planning, ways to support more marginalized populations to access the initiatives etc.). As descriptions of equity considerations may be limited, if there is sufficient data per model, the final product will include a description of equity considerations specific to each model, or actionable strategies that may be relevant for all/most models.

*Types of Scale*: Two researchers with expertise in scaling terminology will review all studies to confirm the listed types of scale (horizontal etc.), with all reviewers and the wider team used as confirmation. The results will be quantified to represent which model was most associated with each type of scale, then presented as a colour coded heat map table. For example, if 95% of studies classified as a hub-and-spoke model were also classified as vertical scale, it would be the strongest colour, while if only 10% of hub-and-spoke models were also classified as spontaneous, it would be much lighter.

## Additional considerations

*Overlapping Authorship*: Given the expertise of the team, co-authors on the review will also be authors of relevant publications. This is an advantage to our work, rather than a conflict of interest, as the authors may provide additional contextual information that will allow us to confirm model names and descriptions.

*Missing Data*: Given that details of the scaling process are rarely provided, there is anticipated to be a lot of missing data. Content analysis and extensive consultation will be used to confirm the list of model names, descriptions, and visual representations that will be used for the review. Models without sufficient evidence for description may be suggested for future exploration but will not be included in the final list of model names. Future work will focus on developing expert consensus on the model names and descriptions.

## Timeline and progress to date

At time of protocol submission, the search strategy was complete. Title/abstract screening has begun and will be followed by full text screening, then data extraction. No data extraction, synthesis, or analytic decisions have been finalized. The protocol has not been modified in response to preliminary findings. Screening is estimated to be completed by mid-2026, data extraction completed by late-2026, aiming for final results in 2027.

## Discussion

This novel review aims to inform practical planning of how to bring healthcare initiatives to new settings and populations. Digital health initiatives are an ideal place to study this concept as there are large investments being made in this area, the building of digital architecture is being done at a large scale providing more examples to study, and digital tools are typically more scalable, making them more likely to be chosen for spread and scale. [36–41] We also anticipate these models may be applicable to other areas of healthcare. A list of structural models for spreading and scaling initiatives in digital health will be developed, including the model names, descriptions, visual representation, strengths, limitations, preconditions, barriers and facilitators to application, and equity considerations. The anticipated result tables will include:

- Summary of metadata (number of studies, countries etc.)
- Summary of intervention studies (type of study, number of sites etc.)
- List of models for spread and scale including name, description and visual representation.
- Table of strengths, limitations, and preconditions for success associated with each model.
- Table of barriers and facilitators to the implementation of each model.
- Table of equity considerations for each model (or overall if there is insufficient data per model).
- Table of associations between models and Types of Scale (horizontal etc.).

A 2023 review by Corôa et al. states that *"More rigorous methodologies and systematic approaches should be encouraged in the science of scaling."* [12] This review is a step towards more rigorous development of structural models of

spread and scale, which aim to support more practical approaches in spread and scale planning, and more consistent terminology. We aim to build off of the work of ExpandNet to connect models with Types of Scale, [31] while also providing more specific structural representations than is provided through the McLean definitions of scaling up, out, and deep. [62] Previous work with non-governmental organizations has classified "pathways to scaling-up", which included quantitative, functional, political, and organizational, and these descriptions are considered in the initial list of models, to see if they are also relevant when considering structural models for spreading and scaling digital health initiatives [63].

Scaling is a complex, non-linear process [3,12,23,24]. Naming structural models of scale may not sufficiently acknowledge this complexity. We also recognize that this review will not identify all models of scale, our naming of models may be subjective due to lack of consistent terminology in the field, and multiple models can be occurring simultaneously, adding further complexity. The complex adaptive nature of the health system also necessitates a 'methodological pluralism' to account for the rich, and sometimes paradoxical, phenomena which regularly impacts the scaling process. Structural models of spread and scale may be a useful starting point for teams who are looking for guidance in the implementation of their digital health initiative. To support more equitable delivery of healthcare services, we need more specific direction on how to plan for sustainable spread and scale so more deliberate planning can occur and continue long-term compared to the current focus on spontaneous and opportunistic approaches, which may end up omitting those most in need of the benefit provided by the initiative. We see these structural models as one piece of a very complex process.

Throughout this review, we aim to build off implementation science literature, while also trying to focus on using language that is easy for all to understand and apply. We will work closely with our partners with lived and living experience to refine the model names and descriptions to align with the literature, while remaining easy to understand. Partners with lived experience are increasingly involved in scale planning [64], and we aim for these models to be understood by all. Future work aims to convert our results, alongside other relevant evidence, into tools that will make these results accessible for: 1) organizations and teams that want to scale their effective initiatives; 2) policy makers who are scaling initiatives; and 3) scientists that can strengthen the science about spread and scale. Future work will also test if/how these models apply to other areas of healthcare.

## Limitations

As there is currently no list of structural models for scale, the search could not focus on papers with named models as this would exclude papers which scaled initiatives but could not name their scaling model. This means there is a necessary level of subjectivity to the analysis, particularly for the inclusion of *author-derived* names of models, rather than relying on model names from the literature (*literature-derived* names). Experts in spread and scale are authors on this protocol and will be included in the discussion about model names and description. We also plan for a necessary next step where we will conduct a consensus process with international experts to verify the names and descriptions, recognizing there will likely be new models suggested. Understanding of these models will deepen with further research, particularly once we have some preliminary names to guide the process. Although we anticipate these results will apply to other areas of healthcare, digital health initiatives provided an ideal area to begin this review, given the rapidly increasing number of digital health initiatives that are or are intending to be scaled. Future research will also be needed regarding face and external validity of the models within and outside of digital health initiatives.

## Conclusion

This novel review aims to name, describe, and visually represent structural models for spreading and scaling digital health initiatives. We will also identify the most consistently important features of each model that need to be considered when planning to bring an initiative to more settings and populations. This novel approach supports more deliberate planning for spread and scale, which has been shown to support more equitable care delivery. In the inconsistent and fragmented field

of scaling healthcare initiatives, this work aims to provide a foundation upon which we can begin to deepen our understanding of how to equitably and sustainably, spread and scale effective initiatives to benefit more people.

## Supporting information

**Appendix 1. Medline Search Strategy.**
(DOCX)

**S1 File. Laur-PRISMA-P-checklist-Oct28,2025.**
(PDF)

## Acknowledgments

The authors wish to thank members of the Office of Spread and Scale, and the Women's College Hospital Research and Innovation Institute, for continually providing feedback as this idea developed.

## Author contributions

**Conceptualization:** Celia Laur, Karen Lee, Andrew Milat, Samuel Petrie, Zeenat Ladak, Alix Hall, Nicole Nathan, Priscilla Medeiros, Noah Ivers, Onil Bhattacharyya.

**Data curation:** Celia Laur.

**Investigation:** Celia Laur, Karen Lee, Andrew Milat, Samuel Petrie, Zeenat Ladak, Alix Hall, Nicole Nathan, Priscilla Medeiros, Noah Ivers, Onil Bhattacharyya.

**Methodology:** Celia Laur, Karen Lee, Andrew Milat, Samuel Petrie, Zeenat Ladak, Vincci Lui, Alix Hall, Nicole Nathan, Priscilla Medeiros, Noah Ivers, Onil Bhattacharyya.

**Project administration:** Celia Laur.

**Supervision:** Celia Laur.

**Writing – original draft:** Celia Laur.

**Writing – review & editing:** Celia Laur, Karen Lee, Andrew Milat, Samuel Petrie, Zeenat Ladak, Vincci Lui, Alix Hall, Nicole Nathan, Priscilla Medeiros, Noah Ivers, Onil Bhattacharyya.

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
