## [Decision Letter · Decision Letter 0]

7 Jan 2026

 If applicable, we recommend that you deposit your laboratory protocols in protocols.io to enhance the reproducibility of your results. Protocols.io assigns your protocol its own identifier (DOI) so that it can be cited independently in the future. For instructions see: https://journals.plos.org/plosone/s/submission-guidelines#loc-laboratory-protocols. Additionally, PLOS ONE offers an option for publishing peer-reviewed Lab Protocol articles, which describe protocols hosted on protocols.io. Read more information on sharing protocols at https://plos.org/protocols?utm_medium=editorial-email&utm_source=authorletters&utm_campaign=protocols. We look forward to receiving your revised manuscript.Kind regards,Udoka Okpalauwaekwe, MD, MPH, PhDAcademic EditorPLOS One  Journal Requirements:

2. We noted in your submission details that a portion of your manuscript may have been presented or published elsewhere.

[The protocol has been submitted for registration with OSF.]

Please clarify whether this publication was peer-reviewed and formally published. If this work was previously peer-reviewed and published, in the cover letter please provide the reason that this work does not constitute dual publication and should be included in the current manuscript.

**Comments to the Author** 1. Does the manuscript provide a valid rationale for the proposed study, with clearly identified and justified research questions? The research question outlined is expected to address a valid academic problem or topic and contribute to the base of knowledge in the field.Reviewer #1: Yes________________________________________2. Is the protocol technically sound and planned in a manner that will lead to a meaningful outcome and allow testing the stated hypotheses? The manuscript should describe the methods in sufficient detail to prevent undisclosed flexibility in the experimental procedure or analysis pipeline, including sufficient outcome-neutral conditions (e.g. necessary controls, absence of floor or ceiling effects) to test the proposed hypotheses and a statistical power analysis where applicable. As there may be aspects of the methodology and analysis which can only be refined once the work is undertaken, authors should outline potential assumptions and explicitly describe what aspects of the proposed analyses, if any, are exploratory.Reviewer #1: Partly________________________________________3. Is the methodology feasible and described in sufficient detail to allow the work to be replicable?  Descriptions of methods and materials in the protocol should be reported in sufficient detail for another researcher to reproduce all experiments and analyses. The protocol should describe the appropriate controls, sample size calculations, and replication needed to ensure that the data are robust and reproducible.Reviewer #1: Yes________________________________________4. Have the authors described where all data underlying the findings will be made available when the study is complete? The PLOS Data policy requires authors to make all data underlying the findings described in their manuscript fully available without restriction, with rare exception, at the time of publication. The data should be provided as part of the manuscript or its supporting information, or deposited to a public repository. For example, in addition to summary statistics, the data points behind means, medians and variance measures should be available. If there are restrictions on publicly sharing data—e.g. participant privacy or use of data from a third party—those must be specified.Reviewer #1: Yes________________________________________5. Is the manuscript presented in an intelligible fashion and written in standard English? PLOS ONE does not copyedit accepted manuscripts, so the language in submitted articles must be clear, correct, and unambiguous. Any typographical or grammatical errors should be corrected at revision, so please note any specific errors here.Reviewer #1: Yes________________________________________6. Review Comments to the Author Please use the space provided to explain your answers to the questions above and, if applicable, provide comments about issues authors must address before this protocol can be accepted for publication. You may also include additional comments for the author, including concerns about research or publication ethics. You may also provide optional suggestions and comments to authors that they might find helpful in planning their study. (Please upload your review as an attachment if it exceeds 20,000 characters)

Reviewer #1: Reviewer comments PONEIt was a pleasure reviewing your protocol which addresses an important gap in implementation and scale-up science by proposing a systematic approach to naming, describing, and comparing structural models used to spread and scale digital health initiatives.Overall, the methods are thorough and well aligned with JBI and PRISMA-ScR methodology. The topic is also timely given the global push toward virtual care and digital health scale-ups post-COVID-19. The writing is clear and organized. However, several aspects require clarification to strengthen reproducibility, transparency, and rigor, particularly around conceptual definitions, positionality of the research team, and the plan for managing anticipated subjectivity in derived naming of models. See comments below:1) How will extractors decide whether a description in a paper constitutes a structural model vs. a scale-up framework or mechanism? And What minimum criteria qualify something as a “model”? I ask because you state that structural models are distinct from process models and theories, yet the distinction remains conceptually fuzzy.2) You mention that because many studies lack explicit model names , names will be derived by reviewers. Don’t you think this introduces interpretative bias? Why not add explanation on how disagreements will be handled (e.g., consensus threshold, third reviewer and state whether authors will record rationales for naming decisions (to enable reproducibility).3) I feel your equity Implementation analysis plan is underdeveloped. You fail to indicate whether communities or people with lived experience were involved in shaping extraction criteria, and whether cultural or structural determinants (e.g., Indigenous sovereignty, rurality, broadband inequity) will be captured.4) You state that the search strategy is already fully completed and title/abstract screening has begun. I wonder if you checked the Plos One requirements for protocols which states that protocols should typically be posted before significant screening occurs. Thoughts?5) Will searches will be limited by geography6) Will only English-language grey literature will be included (main text limits publications by English only; grey lit unclear)7) Your PCC table criteria could acknowledge emerging tech not yet in health systems (e.g., AI imaging triage)8) Your use of duplicate reviewer screening is appropriate, but protocol should specify if Cohen’s statistic or percent agreement will be reported9) Provide justification for your data availability statement.10) Add citation to PRISMA-ScR checklist in main text rather than supplementary only.________________________________________7. PLOS authors have the option to publish the peer review history of their article (what does this mean?). If published, this will include your full peer review and any attached files.  ?>**Do you want your identity to be public for this peer review?**

---

## [Author Response · Author response to Decision Letter 1]

20 Jan 2026

Thank you for reviewing this protocol and providing these useful suggestions. We have made all suggested changes.

1. We note your current Data Availability statement: "N/A - No results are reported-

Given the broad approach of extracting text whose relevance may not be determined until the analysis stage, the data extraction will not be made publicly available. However, other researchers are welcome to contact the team to receive that data, with limitations clearly outlined. Codebooks and any guiding lists of known models of spread and scale and definitions will be made publicly available."

Please note that we recommend the following Data Availability Statement for Study Protocols:

"No datasets were generated or analyzed during the current study. All relevant data from this study will be made available upon study completion.

This change has been made, and the Data Availability statement now reads:

No datasets were generated or analyzed during the current study. All relevant data from this study will be made available upon study completion.

---

## [Decision Letter · Decision Letter 1]

8 Feb 2026

Structural models for spreading and scaling digital health initiatives: A scoping review protocol

PONE-D-25-58053R1

Dear Dr. Laur,

We’re pleased to inform you that your manuscript has been judged scientifically suitable for publication and will be formally accepted for publication once it meets all outstanding technical requirements.

Kind regards,

Udoka Okpalauwaekwe, MD, MPH, PhD

Academic Editor

PLOS One

Additional Editor Comments (optional):

Reviewers' comments:

Reviewer's Responses to Questions

**Comments to the Author**

1. Does the manuscript provide a valid rationale for the proposed study, with clearly identified and justified research questions?

Reviewer #1: Yes

2. Is the protocol technically sound and planned in a manner that will lead to a meaningful outcome and allow testing the stated hypotheses?

Reviewer #1: Yes

3. Is the methodology feasible and described in sufficient detail to allow the work to be replicable?

Reviewer #1: Yes

4. Have the authors described where all data underlying the findings will be made available when the study is complete?

Reviewer #1: Yes

5. Is the manuscript presented in an intelligible fashion and written in standard English?

Reviewer #1: Yes

You may also provide optional suggestions and comments to authors that they might find helpful in planning their study.

Reviewer #1: The authors sufficiently addressed my comments and I have no further comments at this time. Wish you the best

**Do you want your identity to be public for this peer review?** For information about this choice, including consent withdrawal, please see our Privacy Policy

Reviewer #1: No

---

## [Editor Report · Acceptance letter]

PONE-D-25-58053R1

PLOS One

Dear Dr. Laur,

I'm pleased to inform you that your manuscript has been deemed suitable for publication in PLOS One. Congratulations! Your manuscript is now being handed over to our production team.

Kind regards,

on behalf of

Dr. Udoka Okpalauwaekwe

Academic Editor

PLOS One